# The Neuroimmune Regulation and Potential Therapeutic Strategies of Optic Pathway Glioma

**DOI:** 10.3390/brainsci13101424

**Published:** 2023-10-07

**Authors:** Khushboo Irshad, Yu-Kai Huang, Paul Rodriguez, Jung Lo, Benjamin E. Aghoghovwia, Yuan Pan, Kun-Che Chang

**Affiliations:** 1Department of Symptom Research, University of Texas MD Anderson Cancer Center, Houston, TX 77030, USA; kirshad@mdanderson.org (K.I.); beaghoghovwia@mdanderson.org (B.E.A.); 2Division of Neurosurgery, Department of Surgery, Kaohsiung Medical University Hospital, Kaohsiung 80708, Taiwan; yukaih@gmail.com; 3Graduate Institute of Medicine, College of Medicine, Kaohsiung Medical University, Kaohsiung 80708, Taiwan; 4Department of Ophthalmology, Louis J. Fox Center for Vision Restoration, University of Pittsburgh School of Medicine, Pittsburgh, PA 15213, USA; rodriguez.paul@medstudent.pitt.edu; 5Department of Ophthalmology, Kaohsiung Chang Gung Memorial Hospital and Chang Gung University College of Medicine, Kaohsiung 83301, Taiwan; enchantvik@gmail.com; 6Department of Neuro-Oncology, University of Texas MD Anderson Cancer Center, Houston, TX 77030, USA; 7Department of Neurobiology, Center of Neuroscience, University of Pittsburgh School of Medicine, Pittsburgh, PA 15213, USA

**Keywords:** optic pathway glioma, optic nerve, neurofibromatosis type 1, tumor microenvironment, cancer neuroscience, tumor-associated microglia and macrophages, T cells

## Abstract

Optic pathway glioma (OPG) is one of the causes of pediatric visual impairment. Unfortunately, there is as yet no cure for such a disease. Understanding the underlying mechanisms and the potential therapeutic strategies may help to delay the progression of OPG and rescue the visual morbidities. Here, we provide an overview of preclinical OPG studies and the regulatory pathways controlling OPG pathophysiology. We next discuss the role of microenvironmental cells (neurons, T cells, and tumor-associated microglia and macrophages) in OPG development. Last, we provide insight into potential therapeutic strategies for treating OPG and promoting axon regeneration.

## 1. Introduction

Optic pathway glioma (OPG) is a type of brain tumor that develops within the optic pathway, which connects the eye to the brain. Most OPGs (59%) arise in children younger than 10 years of age [1]. Most OPGs are histologically characterized as grade 1 pilocytic astrocytoma (2021 WHO classification), in which *KIAA1549-BRAF*, *BRAF*, and *NF1* are the commonly altered genes [2]. OPGs are low-grade gliomas with features that include a low proliferation index [3,4] as well as glial fibrillary acidic protein (GFAP) and oligodendrocyte transcription factor 2 (OLIG2) immunoreactivity [5,6], and they are often slow-growing. Whereas glial markers in OPG display high positivity, they stain negatively for NeuN, indicating the glial feature exhibited by the tumor cells [5]. OPGs can develop in both anterior (prechiasmatic) and posterior (chiasmatic and postchiasmatic) visual pathways. Because the tumor is located along the optic pathway, patients may experience vision decline or vision loss in severe cases [7,8,9,10]. Some OPGs are located close to the hypothalamus and can thus induce endocrine abnormalities, including precocious puberty [10,11].

Optic pathway gliomas can be categorized into two groups: neurofibromatosis type 1 (NF1)-associated OPG and sporadic (non-NF1) OPG. As an autosomal dominant genetic disease, NF1 is a cancer predisposition syndrome that affects 1 in 2500–3000 people worldwide; 15–20% of children with NF1 develop gliomas, approximately 65–75% of which are OPGs [9,10,12,13]. Children with NF1-OPG tend to develop anterior tumors more frequently than children with sporadic OPG [14,15,16,17]. The spontaneous regression of OPG has been reported in NF1 patients [18,19,20]. In sporadic OPGs, the most common genetic event is *BRAF-KIAA* fusion [21], which increases BRAF activity and its downstream MAPK signaling. NF1-OPG has a better prognosis than sporadic OPG, probably because of the higher incidence of anterior tumors along the optic pathway in NF1-OPGs [10,22].

The *NF1* gene encodes neurofibromin protein, a large protein comprising approximately 3000 amino acids and with partial-sequence homology to a family of proteins that inhibit RAS activity, also known as RAS GTPase-activating proteins (RAS-GAPs) [23]. Neurofibromin can inhibit several RAS proteins—namely HRAS, NRAS, and KRAS—through RAS-GTP hydrolysis (the conversion of active RAS-GTP to inactive RAS-GDP), leading to the abrogation of RAS activity [24,25]. Therefore, it is understandable that in NF1 patients, loss-of-function *NF1* mutations heighten RAS activity, which in turn may be responsible for tumorigenesis [26,27]. Indeed, in both murine and human NF1-associated tumors, reduced neurofibromin expression is accompanied by RAS protein hyperactivity and the elevated activation of the RAS downstream pathways (e.g., RAF-MEK-ERK, PI3K-AKT, mTOR signaling pathways) [28,29,30]. Neurofibromin also has RAS-independent functions: for example, it positively mediates cyclic adenosine monophosphate (cAMP) levels in retinal ganglion cells (RGCs) via PKC-zeta/GRK2 [31].

The study of human OPG is difficult because OPGs are rarely resected, and even when they are, the low-grade glioma cells are rarely grown in culture or as xenografts [32,33]. To overcome this barrier and gain a deeper understanding of OPGs, researchers have generated genetically engineered mouse models of OPG [34,35,36,37,38,39]. One such model, *Nf1*^OPG^ (*Nf1*^flox/−^; Gfap-Cre) mice, recapitulates the genetics of NF1 patients: the mice harbor a germline *Nf1* mutation in all cells and a second-hit *Nf1* loss in the glioma-initiating neural stem cells [34].

## 2. Methods

Following the PRISMA guidelines, authors searched the PubMed/MEDLINE and Web of Science databases and examined the relevant literature published from 1980 to 2023. Combinations of the following keywords were used to screen the topic, maximizing the specificity and sensitivity: “optic pathway glioma”, “neurofibromatosis type 1”, “cancer neuroscience”, “cancer therapy”, and “neuroinflammation”. We used these combination keywords to further screen the title and abstract.

## 3. Results

### 3.1. Effect of Optic Pathway Glioma on Vision and Retinal Ganglion Cell (RGC) Function

Approximately 30–50% of patients with NF1-OPG experience vision decline [13,40,41,42,43]. The radiographic characteristics of the OPG tumor do not seem to correlate with visual outcome [41,44,45], although a recent study suggests that larger OPG volume (measured by magnetic resonance imaging [MRI]) is associated with less retinal nerve fiber layer thickness [46]. Most NF1-OPG patients exhibit good visual outcomes [47]. Conversely, sporadic OPGs have a higher chance of progressing and producing worse visual outcomes than NF1-OPGs [10,22,48,49,50].

Although some support using MRI screening for the early diagnosis of NF1-OPG and, thus, better vision outcomes [51], some may oppose MRI scans of young patients due to the effects of the repeated sedation involved in the process [52]. Sometimes MRI is carried out only when optic nerve abnormalities or visual impairments are detected [47], and MRI may not detect a tumor or tumor progression when vision loss is observed. Optical coherence tomography (OCT) is becoming increasingly popular for monitoring vision and can detect changes in retinal nerve fiber layer thickness early on, thereby facilitating crucial decision-making processes [53,54]. Similarly, the electrophysiological monitoring of OPG patients by using electroretinography (ERG) and visual evoked potentials (VEP) may also provide early evidence of optic nerve or RGC dysfunction [55,56,57]. However, the choice of modality is still at the discretion of the treating clinician. The optimal diagnostic modality for NF1-OPG management may be better clarified in future multicenter trials.

Like patients with NF1-OPG, the *Nf1*^OPG^ mice exhibit thinner retinal nerve fiber layers as measured by OCT and reduced VEP amplitudes than the non-tumor controls [58,59]. The *Nf1*^OPG^ mice also display RGC death and optic nerve myelin defects [58,59]. Some NF1-associated retinal and visual deficits may potentially be intrinsic to the germline *NF1* mutations, because *Nf1*+/− RGCs exhibit more frequent cell death, shorter neurite length, and smaller growth cone area than RGCs from wild-type counterparts [31]. The neurofibromin-cAMP axis is responsible for these phenotypes observed in primary *Nf1*+/− RGC culture [31].

Several risk factors for OPG-induced visual impairment have been identified, including age <2 years, postchiasmatic tumor location, and female sex [9,10,17,60]. In patients with NF1-OPG, females are more likely than males to undergo MRI and treatment for visual symptoms. Similarly, in mouse models of *Nf1*^OPG^, females have more severe RGC loss than males [42].

### 3.2. The Role of Immune Cells in Optic Pathway Glioma Growth

Microglia and macrophages are the immune cells found in the central nervous system (CNS). However, the origins of microglia and macrophages are different. Microglia are derived from yolk sac progenitors and are the resident immune cells in the brain, whereas macrophages are derived from circulating monocytes and can infiltrate the CNS from the bloodstream [61]. Microglia are important for maintaining brain homeostasis and responding to various types of insults, including pathogens [62,63], injury [64], and neurodegeneration (e.g., Alzheimer’s disease) [65,66]. In the context of brain tumors, microglia and macrophages can contribute to tumor progression.

Glioma cells can recruit and reprogram microglia and macrophages to adopt a pro-tumor phenotype, collectively known as glioma-associated microglia and macrophages (GAMs) [67]. GAMs can release growth factors, cytokines, and chemokines that promote tumor cell survival, invasion, and angiogenesis [68]. Targeting GAMs has been suggested as a potential therapeutic strategy for gliomas [69,70]. Overall, microglia and macrophages are critical components of the CNS immune system and have complex roles in brain tumor biology.

Although both macrophages and microglia exist in the optic nerve [71], microglia are often reported to be involved in several optic neuropathies, including optic nerve trauma [72] and OPGs [73]. Growing evidence shows that the inhibition of microglia activation by the c-Jun-NH(2)-kinase (JNK) inhibitor [74] or minocycline (a crude microglia inhibitor) [75,76] is able to delay glioma formation in the *Nf1*^OPG^ mice. In addition to pharmaceutical approaches, genetically reducing the expression of CX3CR1, the receptor that modulates microglial migration [77], delays OPG formation [73]. Among the secretion molecules from GAMs, chemokine (C-C motif) ligand 5 (CCL5) is a key promoter for NF1-OPG growth. Neutralization with CCL5 antibody [78] or the deletion of *Ccl5* [79] dramatically attenuated tumor growth in vivo. An interesting observation suggests that the reduced RGC numbers and thinner retinal nerve fiber layer in female *Nf1*^OPG^ mice compared with male ones may be due to estrogen-induced microglial activation [80]. Thus, suppressing microglial activation, either during cell migration or mitogen secretion, is a potential therapeutic strategy against OPG.

T cells (T lymphocytes) are white blood cells that modulate adaptive immune response [81,82]. T cells are further identified as CD4+ helper T cells and CD8+ cytotoxic T cells [83]. In cancer pathology, CD8+ T cells are detected around the tumors and are thought to mediate anti-tumor responses [84]. However, CD8+ T cells fail to do so during long-period tumorigenesis because they differentiate into dysfunctional states [85]. In the case of mouse NF1-OPG, CD8+ T cells are involved in neuron-mediated T cell CCL4 production and subsequently induce microglial CCL5 secretion to support tumor growth, and the depletion of CD8+ T cells suppressed tumor growth [86], thereby suggesting that CD8+ T cells could be a therapeutic target in the optic nerve for patients with OPG.

### 3.3. The Role of Neurons in Optic Pathway Gliomagenesis

In the past years, ample studies have confirmed the key function of neurons in modulating cancer progression [87]. OPG grows within the optic pathway and is surrounded by neurons. Preclinical studies have shown that light-induced neuronal activity in the optic nerve is required to initiate NF1-OPG. In the *Nf1*^OPG^ mice, dark-rearing inhibits tumor initiation and maintenance [88]. Molecularly, the germline *Nf1* mutation aberrantly increases neuronal activity-induced secretion of ADAM10, a protease that cleaves membrane proteins, in the optic nerve. One of the substrates of ADAM10 is neuroligin-3. This *Nf1*/ADAM10 axis results in the increased cleavage of the ectodomain of neuroligin-3, which is sufficient for increasing OPG cell growth in vitro. Supporting the role of ADAM10/neuroligin-3 axis in the pathogenesis of NF1-OPG, genetically ablating neuroligin-3, or pharmacologically inhibiting ADAM10 in *Nf1*^OPG^ mice inhibits optic gliomagenesis [88].

In addition to the light-dependent regulation of NF1 optic gliomagenesis, a light-independent neuronal pathway regulates OPG growth [89]. Certain germline *Nf1* mutations in RGCs induce neuronal hyperactivity by inhibiting the hyperpolarization-activated, cyclic nucleotide-gated (HCN) channels. This RGC hyperactivity increases neuronal production of midkine, which functions to recruit T cells that support OPG growth, as discussed in the above section. Midkine-neutralizing antibodies rescued the *Nf1*-mutation-induced RGC hyperactivity and inhibited tumor growth in the *Nf1*^OPG^ mice [89].

The above-mentioned studies were conducted in the context of NF1-OPGs. Whether neurons are important microenvironmental contributors to sporadic OPGs that do not harbor the germline *NF1* mutation remains to be determined.

### 3.4. Implications for Optic Pathway Glioma Treatment

#### 3.4.1. Targeting the Neoplastic Cells

Because surgery may damage the optic pathway and radiation imposes risks for secondary malignancies and cognitive issues in NF1-OPG, the first line treatment for NF1-OPG is chemotherapy, which often involves platinum-based and vinca alkaloids-based chemotherapy agents (e.g., carboplatin and vincristine) that target the neoplastic (tumor) cells [13,90,91,92,93,94,95]. Because both NF1-associated and sporadic OPGs exhibit MAPK hyperactivation, targeted therapy is a potential strategy. In addition, *NF1* loss leads to the hyperactivation of RAS and its downstream effectors (e.g., MEK/ERK, PI3K/AKT, and mTOR). Targeting these pathways inhibits tumor growth in preclinical mouse models [30]. Ongoing clinical trials primarily aim to target these aberrantly hyperactive pathways within the neoplastic cells [96,97].

#### 3.4.2. Targeting the Microenvironment

Given the emerging evidence that the microenvironment (e.g., neurons, T cells, microglia) is critical for OPG pathogenesis, future clinical studies could consider adjuvant strategies that target microenvironmental factors, including ADAM10, neuroligin-3, midkine, CCL4, and CCL5. Environmental factors, especially light, could potentially serve as targets for preventing or treating OPG; however, because light is important for the development of visual pathway, preclinical studies should be conducted to determine the optimal light conditions that can be used to treat OPG without affecting visual pathway development. In addition, ADAM10 inhibitors and HCN activators may represent new neuron-targeting strategies to treat OPG. In malignant high-grade gliomas, neuronal activity leverages the ADAM10/neuroligin-3 axis to drive tumor growth [98,99], which has led to a clinical trial that evaluates an ADAM10 inhibitor (INCB7839) in treating recurrent or progressive pediatric high-grade gliomas (NCT04295759). Lamotrigine, an HCN agonist that is used clinically for treating seizures, rescues *Nf1*-mutation–induced RGC hyperactivity and inhibits tumor growth in *Nf1*^OPG^ mice.

In addition to targeting the neuronal components, targeting the tumor immune microenvironment is another attractive strategy. Tumor associated microglia and macrophages (TAMs) and myeloid-derived suppressor cells (MDSCs) are tumor-promoting in many solid tumors [100,101]. Because TAMs are the most abundant immune cells in the tumor microenvironment and can be associated with poor prognosis of patients [102], several small-molecule inhibitors have been developed against TAMs and their inflammatory signaling (e.g., in prostate, breast, ovarian, colon, and skin cancers) [103]. Although successful cases of tumor inhibition were reported by small molecule treatment, liver and renal toxicities remain issues [104,105].

The development of a TAM-targeting strategy for treating OPG remains an unmet need. In preclinical models, CD8+ T cells were shown to secrete CCL4, which binds to microglial CCR5 to stimulate the microglial secretion of CCL5 via the nuclear factor-κB (NFkB) pathway [86]. Targeting these critical molecules in the immune regulation of OPG may be an effective strategy. Additionally, using a natural small molecule to inhibit aldose reductase was found to suppress microglia and macrophage migration and inflammatory cytokine secretion in mouse eyes [106,107]. Studies also showed that aldose reductase inhibition alleviated microglia activation triggered by beta-amyloid [66] and optic nerve injury [108], suggesting that aldose reductase blockade plays a protective role in the CNS and may be a therapeutic avenue for OPG. Table 1 is presented below.

## 4. Discussion

Several rodent models were established for OPG studies (Table 1), in which the tumor can be observed as early as two months of age. However, the study of OPG in larger animal or non-human primates remains an unmet need, despite the recent development of NF1 pig models [110,111] Whereas survival for patients with OPG is excellent, no effective treatments for OPG-induced visual impairment are currently available [8,41,49,50,112,113], likely because tumor-induced RGC loss cannot be reversed. Strategies that improve RGC survival and regeneration and that promote the remyelination of RGC axons are needed to rescue OPG-impaired vision.

Since the molecules/proteins in Table 2 are known activators for OPG pathophysiology, the pharmaceutical development of agonists/antagonists to those molecules may be beneficial for OPG therapy. Interestingly, neuroinflammation has been reported to promote axon regeneration [114], with the chemokine CCL5 seen as a critical factor [115]; however, CCL5 is also a mitogen that increases OPG growth. The balance of neuroinflammation in axon regeneration and glioma growth is another key consideration in treating OPG. In addition, the cell-intrinsic deficits induced by germline *NF1* mutations (e.g., in neurons and oligodendroglial cells) will need to be considered as these strategies are tested in the context of NF1-OPG.

Glioma formation in the optic nerve causes axon degeneration [46], leading to vision impairment. Thus, promoting axon regeneration is one of the goals for treating OPG. In optic neuropathy, adeno-associated virus (AAV)-mediated gene therapy that targets RGC is promising in axon regeneration and partial visual restoration in optic nerve crush models [116,117]. Recently, the food and drug administration (FDA) approved Luxturna, an AAV-based gene therapy for treating RPE65-positive retinal dystrophy [118], which marked a clinical milestone. However, so far, no evidence shows that AAV-mediated gene therapy is able to extend regenerative axons from RGCs to go through OPG in animal models. More studies are required to warrant this therapeutic strategy in treating OPG.

In OPG patients, RGCs die followed by axon degeneration. Unfortunately, RGCs do not regenerate or are not replaced after injury in humans. RGC transplantation seems a feasible approach to replace RGCs, but the source of primary human RGC is limited. Stem cell (SC)-derived RGCs provide the opportunity for cell replacement therapy. In addition to replacing dead cells, SC-derived RGCs also improve endogenous RGC survival [119,120]. A study in non-human primates showed that axons from donor RGCs were observed in the brain [121], suggesting that SC-derived RGC transplantation may restore the axon loss in the optic nerve with OPG. Whether the microenvironment of OPG in the optic nerve would affect donor RGC survival and their axon growth would be an interesting direction for future study.

Radiation therapy is one of the most common treatments against tumors. Due to the location of the OPG, radiation therapy seems to be a better approach than the surgical procedure to eliminate a tumor in the optic nerve. However, several radiation-induced side effects are detrimental (especially for patients with NF1), including visual disturbance (7–17% of patients who received radiation therapy), vasculopathy (higher incidence of vasculopathy in patients <5 years than those >5 years after radiation therapy, 12.5% vs. 3.8%), endocrine deficiency (growth hormone deficiency is the most frequent, 59%), neurocognitive impairment, and secondary malignancy [122].

Bevacizumab (Avastin), which inhibits vascular endothelial growth factor (VEGF) from binding to its receptors, is an anti-cancer drug considered for treating glioblastoma (high-grade glioma) [123] and ovarian cancer [124]. A study showed that bevacizumab may improve vision in OPG patients [125]; yet the role of VEGF in OPG progression and OPG-induced visual impairment remains to be determined.

## 5. Conclusions

Combinatory therapies of chemotherapy, gene therapy, and stem cell replacement therapy may be required to reduce tumor size and rescue vision. Given that RGC rescue strategies may increase the optic nerve neuronal activity that induces OPG growth [88,89] and that some of the gene therapies are based on the deletion of an anti-tumor gene (e.g., *PTEN*) [126,127], the careful optimization of the treatment regimen (e.g., dose, sequence, cell-specificity, and timing) will be required for the translational study. Although curing OPG is full of challenges, several groups in the fields of visual sciences, stem cell biology, and brain tumor biology are taking on bold research initiatives to develop collaborative approaches for treating and/or slowing down the progression of OPG. Any successful preclinical and/or clinical studies will help move the field of pediatric oncology forward and give hope to patients who suffer from vision impairment caused by OPG.

## Figures and Tables

**Table 1 brainsci-13-01424-t001:** Genetically engineered mouse models used to study optic glioma.

Genotype	Optic Glioma Phenotype	Ref
*Nf1*^flox/−^; hGfap-Cre *Nf1*^flox/flox^; hGfap-Cre	tumor developed by 2 months of age	[35]
*Nf1*^flox/neo^; Gfap-Cre *^#^	tumor developed by 3 months of age	[34]
*Nf1*^flox/neo^; *Pten*^flox/+^; Gfap-Cre	tumor developed by 3 months of age	[37]
*Nf1*^flox/neo^; Olig2-Cre	tumor developed by 6 months of age	[109]
*Nf1*^flox/neo^; Prom1-CreER	tumor developed by 3 months of age	[109]
*Nf1*^flox/R681X^; Gfap-Cre	tumor developed by 3 months of age	[39]
*Nf1*^flox/G848R^; Gfap-Cre	no tumor detected by 3 months of age	[39]
*Nf1*^flox/C383X^; Gfap-Cre	25% mice developed tumor by 3 months of age	[36]
*Nf1*^flox/R1278P^; Gfap-Cre	tumor developed by 3 months of age	[36]

* neo, engineered neomycin insertion to disrupt *Nf1* function. ^#^ the activity of the Gfap promoter was detected around E15, whereas the activity of the hGfap promoter was detected around E12.

**Table 2 brainsci-13-01424-t002:** Critical cell/tissue types and the corresponding molecules in optic glioma pathophysiology.

Cell/Tissue Type	Molecule	Role in Optic Glioma Progression	Ref
Retinal ganglion cells	Midkine	stimulate T cell Ccl4 production by binding to LRP1	[86]
Retinal ganglion cells	HCN	*Nf1* mutation induced HCN dysfunction induces neuronal hyperactivity; modulates midkine level	[89]
Optic nerve	ADAM10	increased secretion in response to light-induced neuronal activity; cleaves neuroligin-3	[88]
Optic nerve	Neuroligin-3	required for optic gliomagenesis	[88]
CD8+ T cells	CCL4	stimulate microglial Ccl5 production by binding to CCR5	[86]
Microglia	CCL5	binds to CD44 on tumor cells and increases tumor cell survival	[86]
Microglia	CX3CR1	required for tumorigenesis	[73]

## Data Availability

Not applicable.

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
