# Peer review of "The Neuroimmune Regulation and Potential Therapeutic Strategies of Optic Pathway Glioma"

_brainsci, 2023, doi:10.3390/brainsci13101424_

Round 1

Reviewer 1 Report

Nice article and interesting topic on immunomodulation of optic glioma

I would suggest some comments.

  1) please provide the new WHO classification to define this type of glioma (Iegrade1 and not I)

2) a systematic review with descriptive analysis according to the PRISMA guidelines could be desirable considering the level of reviews currently available in the literature. Just to be a reliable review, this concept could be made clear with a section on the method and results.
3) there is very little discussion and the literature review needs to be discussed better

Minor editing of English is required

Author Response

1) please provide the new WHO classification to define this type of glioma (Iegrade1 and not I)
Response: We thank the reviewer’s comment. We have revised it.2) a systematic review with descriptive analysis according to the PRISMA guidelines could be desirable considering the level of reviews currently available in the literature. Just to be a reliable review, this concept could be made clear with a section on the method and results.

Response: We thank the reviewer’s excellent point. We have revised it accordingly.3) there is very little discussion and the literature review needs to be discussed better

Response: We thank the reviewer’s comment. We have now included two tables and extended the discussion.

Reviewer 2 Report

The present review summarized the application of therapies targeting the neuroimmune regulation of optic pathway glioma. Though brief, this manuscript is understandable and well-organized. However, it would be much appreciated if the authors considered adding a summary of all the aspects discussed and one or two tables centralizing the main findings based on the research (murine models or humans). This addition will significantly improve the quality of the manuscript. The Discussion section could be expanded (at least 3-4 paragraphs). In the present form, it is much like a Conclusion section. Please also create a Conclusions section.

Author Response

The present review summarized the application of therapies targeting the neuroimmune regulation of optic pathway glioma. Though brief, this manuscript is understandable and well-organized. However, it would be much appreciated if the authors considered adding a summary of all the aspects discussed and one or two tables centralizing the main findings based on the research (murine models or humans). This addition will significantly improve the quality of the manuscript. The Discussion section could be expanded (at least 3-4 paragraphs). In the present form, it is much like a Conclusion section. Please also create a Conclusions section.

Response: We thank the reviewer’s comment. We have now included two tables and extended the discussion. A conclusion section is also provided.

Round 2

Reviewer 1 Report

The manuscript provided the required changes

Reviewer 2 Report

I have no further comments.